# A Trilinear Model for the Load–Slip Behavior of Headed Stud Shear Connectors

**DOI:** 10.3390/ma16031173

**Published:** 2023-01-30

**Authors:** Hao Meng, Wei Wang, Rongqiao Xu

**Affiliations:** 1Center for Balance Architecture, Zhejiang University, Hangzhou 310007, China; 2Department of Civil Engineering, Zhejiang University, Hangzhou 310058, China

**Keywords:** headed stud shear connector, beam on the foundation, trilinear model

## Abstract

Headed stud shear connectors are most broadly applied in various composite structures. There exist plenty of empirical formulae for load–slip curves. However, most of them are fitting formulae in particular forms. Due to the lack of physical model support, fitting empirical formulae apply only to cases with similar parameters to the tests. Therefore, this paper analyzes the load–slip curves of existing headed stud connectors, proposes three stages of slip deformation in the shear connectors and the corresponding trilinear model, and presents the analytical formulae for the stiffness and strength of headed stud shear connectors. Firstly, we model the headed studs and surrounding concrete as beams on the foundation model, derive the equivalent shear stiffness equations for headed studs, and establish the load–slip behaviors for the first two stages. Then, the connectors’ shear stiffness and shear strength in the third stage are derived based on the head stud’s plastic deformation characteristics and failure mode. Finally, the numerical results are presented and verified with the existing test results, showing that the trilinear model is conceptually straightforward, easy to apply, and has sufficient accuracy.

## 1. Introduction

Steel–concrete composite structures can reduce the usage of building materials and facilitate assembly construction, which are environment-friendly, low-carbon structures. However, the bond strength between the steel and concrete interface is relatively weak; mechanical shear connectors are necessary to ensure that steel and concrete function together. Thus, shear connectors significantly influence the performance of composite structures. Due to the limited stiffness of shear connectors, relative slip always occurs on the stee–concrete interface during the deformation of composite structures; therefore, load–slip behavior emerges as a critical scientific issue in composite structures.

Headed stud shear connectors have been universally applied in various steel–concrete composite structures due to their advantages, such as superior shear strength and ductility, non-directional shear resistance, and straightforward construction. Extensive push-out tests and numerical simulations have been conducted to study headed stud shear connectors’ mechanical properties thoroughly. For example, Viest [1], Ollgaad et al. [2], Oehlers & Johnson [3], and Hiragi et al. [4] identified two typical failure modes of headed stud shear connectors using push-out tests: stud shank failure and concrete failure. Moreover, they determined the critical parameters affecting the shear behavior of the shear connectors based on parametric analysis and proposed empirical formulae for headed stud shear connectors’ shear strength and load–slip curves. These previous studies presented easy-to-use analytical theories and calculation methods, which established the basis for the widespread application of headed stud shear connectors. However, the failure modes, critical influence parameters, and load–slip curves proposed in the previous studies have apparent limitations due to the restriction of test conditions, structural forms, and material types and properties. As the investigation progressed, researchers found that the concrete properties, stud dimensions, studs arrangement, service conditions, and other aspects significantly influence the shear performance of the headed stud shear connectors, which caused the original empirical formulae to no longer be applicable. Okada et al. [5] and Smith & Couchman [6] analyzed the effect of the longitudinal spacing of the stud group on the shear behavior and found that the shear behavior of the connectors decreases as the longitudinal spacing of studs reduces. The reason is that the core compressive concrete of the connector is disturbed by the deformation of the adjacent stud, which makes the concrete quit working prematurely. Concrete failure occurs in the connectors with too small longitudinal spacing because the shear connectors cannot fully utilize the stud strength. The longitudinal spacing affects the load–slip behavior and makes the traditional empirical formulae inapplicable. In addition, Badie et al. [7,8], Wang et al. [9], Lee et al. [10], and Nguyen & Kim [11] found that large-diameter studs’ failure modes and shear behavior differ significantly from conventional-size studs. The reason is that normal strength concrete (NSC) is not sufficiently wrapped for large-diameter studs with diameters of 27-32 mm. The original empirical formulae hardly reflect the interaction mechanism between stud and concrete, so they cannot accurately predict the large-diameter connectors’ stiffness, shear strength, and load–slip curves. Chuah et al. [12], An & Cederwall [13], Nie et al. [14], Hegger et al. [15,16], Döinghaus et al. [17], and Wang et al. [18] investigated the effects of high-strength concrete and high-strength steel commonly used in engineering on the shear behavior of headed stud shear connectors. The studies showed that there exists a critical value in the effect of the compressive strength of concrete on the shear behavior. When lower than this critical value, increasing the compressive strength of concrete can significantly enhance the shear behavior of connectors; when higher than this critical value, the compressive strength of concrete has no apparent effect on the shear behavior. Unlike NSC stud connectors, studs embedded in high-strength concrete (HSC) are subject to higher local support strength, resulting in stud connectors failing with much more shearing than bending, so the former empirical formulae are no longer appropriate for the HSC situation. Kim et al. [19], Cao et al. [20], Kruszewski et al. [21], and Hu et al. [22] conducted an experimental study on headed studs in ultra-high performance concrete (UHPC). It was found that the high compressive strength and ultra-ductile properties of UHPC can provide adequate support and wrapping to the studs, causing the failure mode to pure shearing and UHPC to remain intact, significantly different from the connectors in NSC. Therefore, the critical parameters of the shear behavior for headed stud shear connectors in NSC are no longer evident, including a slender ratio (the stud height to diameter), concrete Youngs’ Modulus, and concrete compressive strength. However, the dimension of the welding foot and the welding material property becomes prominent in UHPC, which less impacts the shear behavior in NSC. The main design objective of the connectors also shifted from shear stiffness and strength design for studs in NSC to the ductility design for UHPC. Apparently, the original empirical formulae cannot reasonably predict the headed stud shear connector’s shear behavior in UHPC. It should be noted that the tension behavior of headed stud shear connectors plays a significant role in the bearing behavior of composite structures [23,24]. The headed stud shear connectors must resist uplift forces in addition to shear forces when composite structures experience complex loading or large deformation conditions. Therefore, studies have been carried out to determine the effects of tension and combined tension–shear loads on headed studs; various empirical relationships have been proposed to estimate the relationship between shear and tensile resistance of headed studs [25,26,27,28].

The above analysis shows that since the connectors’ material and geometric parameters and service conditions are changed, the load–slip curve, shear stiffness, shear strength, and other empirical formulae obtained by fitting test data are no longer applicable. We can indeed still conduct push-out tests on specific headed stud connectors and then obtain the new fitting formulae or adjust the relevant parameters in the conventional ones according to the test results. However, almost all the current load–slip curve formulae are obtained by fitting push-out test results and adjusting the original coefficients from the early two types of empirical formulae: the exponential and fractional forms. Due to the lack of a mechanical model, the exponential and fractional formulae are only the shape-fitting of the load–slip data of the particular test, making it hard to realistically reflect the interaction mechanism of shear connectors and concrete factually. The head stud shear connectors’ shear strengths are also required before applying these formulae. However, the shear strength is associated with the interaction of the stud and concrete. The existing shear strength formulae [29,30,31] only consider two failure modes: stud and concrete failure, which cannot effectively examine the interaction mechanism between the stud and concrete. In summary, the current load–slip curves of the headed stud shear connectors still rely seriously on the push-out test.

On the other hand, a few researchers attempted to investigate the shear resistance of headed stud shear connectors using analytical methods. Tong et al. [32] analyzed the stiffness of headed stud shear connectors in high-strength steel (HSS)–UHPC composite structures based on the beam on Winkler foundation model and derived the analytical stiffness formula. Hu et al. [33,34] proposed a theoretical shear strength formula for headed stud shear connectors in UHPC based on the beam on Winkler foundation model. It is evident that the Winkler foundation beam model can reflect the force transfer mechanism of shear connectors more accurately and offer broader parameter applicability and better expandability theoretically. Meng et al. [35] analyzed the load–slip behavior of headed stud shear connectors in existing tests and developed an entire load–slip curve for headed stud shear connectors based on the beam on Winkler foundation model and the stress–strain relationship of the concrete and stud, and first proposed the load–slip curve formula based on the mechanical model.

Based on the failure modes of headed stud shear connectors and the beam on Winkler foundation model, this paper studies the shear behavior of the shear connectors. The effects of material and geometric parameters of studs, concrete material parameters, and materials degradation on the connectors’ shear behavior are revealed. The analytical trilinear load–slip curve for headed stud shear connectors is proposed and verified using the existing test results, which shows that the model possesses a definite physical sense, is easy to use, and is a powerful complement to the existing research tools.

## 2. Analytical Trilinear Model

### 2.1. Three Stages and a Trilinear Model of the Load–Slip Behaviors

According to many experimental and numerical results in published work, the typical load–slip curve of a headed stud shear connector is schematically shown as a solid line in Figure 1. This paper approximatively describes the load–slip behaviors of the shear connectors as a trilinear model shown as a dashed line plotted in Figure 1. In stage 1, the headed stud and the surrounding concrete are assumed to be in the inelastic range. In contrast, stage 2 corresponds to the stud beginning to yield at its root and the surrounding concrete becoming cracked and degraded in stiffness. In stage 3, a plastic zone forms at the stud root, and the surrounding concrete is degraded with negligible stiffness.

### 2.2. The Quantitative Properties of the First Two Stages

To quantitatively describe the properties of the load–slip behaviors, the headed stud is modeled using the beam. At the same time, the surrounding concrete is idealized using the Winkler foundation, whose foundation stiffness differs between stages 1 and 2. The deformation and the analytical model of the headed stud are schematically plotted in Figure 2.

If the interfacial slip between the concrete slab and the steel beam is Δ, the deflection of the headed stud in the *z*-direction can be expressed by Equation (1), according to the beam on Winkler foundation theory:(1)$$w=C_{2}\cosh\left(ax\right)\sin\left(ax\right)+C_{3}\sinh\left(ax\right)\cos\left(ax\right)+C_{4}\sinh\left(ax\right)\sin\left(ax\right),$$
where:(2)$$\begin{array}{l} C_{2}=-C_{3}=-\frac{\cos\left(ax\right)\sinh\left(ax\right)+\cosh\left(ax\right)\sin\left(ax\right)}{\cos^{2}\left(ax\right)+\cosh^{2}\left(ax\right)-2}\Delta \\ C_{4}=-\frac{2\sin\left(ax\right)\sinh\left(ax\right)}{\sin^{2}\left(ax\right)-\sinh^{2}\left(ax\right)}\Delta \end{array},$$
and:(3)$$a=\sqrt[{4}]{k/\left(4EI\right)},$$
where EI is the bending stiffness of the stud and k denotes the equivalent foundation stiffness of the surrounding concrete idealized using the Winkler foundation. The deflection equation shown in Equation (1) should satisfy the zero rotation condition at both ends of the stud. If the discrete springs of the shear connectors are regarded as a shear spring with a stiffness K among composite members, according to the principle of equivalency of the strain energy, we have:(4)$$\frac{1}{2}K\Delta^{2}=\frac{1}{2}\int_{0}^{h}EI{\left(\frac{d^{2}w}{dx^{2}}\right)}^{2}dx+\frac{1}{2}\int_{0}^{h}kw^{2}dx.$$

Substituting Equation (1) into (4) and eliminating Δ, the shear stiffness K in stages 1 and 2 are:(5)$$K=A_{1}ka^{-1}/16+A_{2}EIa^{3}/4,$$
where $A_{1}$ and $A_{2}$ are the parameters determined by Equations (6) and (7), respectively, in order to simplify the expression of shear stiffness K, where:(6)$$A_{1}=2\left(2s_{2}-2s_{1}+c_{1}s_{2}-s_{1}c_{2}\right)C_{2}^{2}+4\left(c_{2}-s_{1}s_{2}-c_{1}\right)C_{2}C_{4}+\left(2s_{2}+2s_{1}+c_{1}s_{2}-s_{1}c_{2}-2ah\right)C_{4}^{2}$$
(7)$$A_{2}=2\left(2s_{2}-2s_{1}+s_{1}c_{2}-c_{1}s_{2}\right)C_{2}^{2}+4\left(s_{1}s_{2}+c_{2}-c_{1}\right)C_{2}C_{4}+\left(2s_{2}+2s_{1}+c_{1}s_{2}+s_{1}c_{2}+4ah\right)C_{4}^{2}$$
and:(8)$$c_{1}=\cos\left(2ah\right),\;c_{2}=\cosh\left(2ah\right),\;s_{1}=\sin\left(2ah\right),\;s_{2}=\sinh\left(2ah\right).$$

Equation (5) shows the analytical expression of the shear stiffness for the headed stud shear connectors in stages 1 and 2. As long as the stud’s geometrical dimensions, Yang’s modulus, and the equivalent foundation stiffness are known, the shear stiffness can be determined.

### 2.3. The Equivalent Stiffness of the Foundation Idealized from the Surrounding Concrete

The material properties of the concrete significantly influence the load–slip characteristics of the headed stud connectors, which is simplified to the equivalent foundation stiffness k in the present model. Based on the previous derivation, once the equivalent foundation stiffness of the concrete is determined, the stiffness of the headed stud connectors in the first and second stages can be quantified directly using the geometry and material parameters of the connectors. Therefore, determining the equivalent foundation stiffness for concrete becomes a vital issue.

In fact, the interaction between laterally loaded piles and foundation soils in geotechnical engineering is analogous to that between headed studs and concrete. Many researchers have studied the equivalent stiffness for the interaction between various soil types and piles. They made it clear that the equivalent stiffness is related to the material properties of the foundation soil, pile diameter, pile cross-section shape, and the stiffness and strength ratio of the pile to the soil. In addition, researchers have summarized the equivalent foundation stiffnesses for various piles and foundation soils using numerous tests and have tabulated commonly used equivalent foundation stiffnesses for convenience. However, due to the significant differences in material properties between concrete and foundation soils in geotechnical engineering, material and geometric parameters between the pile and headed stud connectors, and Young’s modulus ratio between soil–pile and concrete–stud, the equivalent stiffness formulae established in geotechnical engineering are inapplicable to the case of headed stud shear connectors.

Researchers have provided equivalent foundation stiffness formulae using experimental [36,37] and 3D FEM [35] approaches. However, the dispersion between the formulae is significant. The existing push-out test results of the headed stud shear connectors verify the formula $k={1.5E/d}^{0.5}$ by Meng et al. [29]. However, the other two formulae are not validated using the push-out test for shear connectors. Therefore, we select $k={1.5E/d}^{0.5}$ as the equivalent foundation stiffness of concrete and studs, which can be obtained from the diameter of the studs and Young’s modulus of concrete.

The dispersion between the above equivalent foundation stiffness formulae is significant. Equation (11) is verified using the current push-out test results of the headed stud shear connectors, which is more applicable to this paper. Therefore, we select $k={1.5E/d}^{0.5}$ as the equivalent foundation stiffness of concrete and studs, which can be obtained from the diameter of the studs and Young’s modulus of concrete.

### 2.4. The Degradation of the Surrounding Concrete

According to the assumption of this paper, the shear stress of the stud’s root reaches the yielding limit in stage 2. Therefore, the shear force of the stud at the end of stage 1 is:(9)$$P_{1}=\frac{\sqrt{3}}{12}f_{y}\pi d^{2},$$
where $f_{y}$ denotes the yielding limit of the stud material. Within the first stage, only a few minor cracks appear in the concrete, the whole remains intact, and the load-bearing behavior of the connector exhibits linear elastic characteristics. The concrete develops penetration cracks after the load–slip curve enters the second stage, leading to its stiffness degradation. In this paper, the degradation process is simulated using the reduction of Young’s modulus. The reduced Young’s modulus of the concrete in stage 2 is taken as:(10)$$E_{c}^{II}=\xi E_{c},$$
where ξ is the reduction coefficient, and given by:(11)$$\xi =\rho_{c}n(n-1)(1-{\bar{\varepsilon }}^{n}){\left(n-1+{\bar{\varepsilon }}^{n}\right)}^{-2},$$
in which:(12)$$\rho_{c}=\frac{f_{c,r}}{E_{c}\varepsilon_{c,r}},n=\frac{E_{c}\varepsilon_{c,r}}{E_{c}\varepsilon_{c,r}-f_{c,r}},\bar{\varepsilon }=\frac{\varepsilon }{\varepsilon_{c,r}},$$
where $f_{c,r}$ is the uniaxial concrete compressive strength, while $\varepsilon_{c,r}$ is the strain corresponding to uniaxial compressive strength $f_{c,r}$. ε denotes the compressive strain. 

Substituting the reduced modulus of the concrete in stage 2 into $k={1.5E/d}^{0.5}$ yields the equivalent foundation parameter. Then, the shear stiffness of the connector in the second stage is obtained by substituting Equation (10) into Equation (5).

### 2.5. Stage 3 of the Load–Slip Curve

With the slip increase, the concrete’s Young’s modulus gradually reduces until the concrete around the stud root completely quits working. At the same time, the plastic zone of the stud at its root enters the plastic-hardening phase. As Figure 3 shows, the surrounding concrete’s failure mode is assumed to be a shear failure.

Meng et al. [35] and Xu et al. [38] investigated the height of the concrete plastic zone using the analytical approach and the 3D FEM, respectively. The comparison between the concrete plastic zone d assumed in the present model and those provided by the existing literature are shown in Table 1. It is not hard to find that the ratio between the stud diameter and the equivalent length of the concrete plastic zone in the existing study is close to 1. Therefore, the stud diameter d is taken as the characteristic height of the concrete plastic zone in this paper.

The tension strain of the concrete is given by:(13)$$\varepsilon_{c}=\frac{\sqrt{d^{2}+{\left(L+\Delta \right)}^{2}}-\sqrt{d^{2}+L^{2}}}{\sqrt{d^{2}+L^{2}}},$$

Consequently, the relative slip $\Delta_{2}$ at the end of stage 2, corresponding to the ultimate limit of the surrounding concrete is:(14)$$\Delta_{2}=\sqrt{L^{2}+2\varepsilon_{c,u}L^{2}+2\varepsilon_{c,u}d^{2}}-L,$$
where L denotes the longitudinal space of the stud and L=5d for L≥5d, and $\varepsilon_{c,u}$ is the nominal limited strain.

Since the slip is larger than $\Delta_{2}$, the load–slip behavior enters stage 3. The concrete quits working; that is to say, the stiffness of the shear connector entirely comes from the stud experiencing plastic hardening. As shown in Figure 4, assuming the plastic zone at the stud root is experiencing pure shearing, with the plastic zone height of d, and the plastic hardening rule for headed studs is linear hardening, the shear force is:(15)$$P_{sp}=G_{sp}\mathrm{atan}(\frac{\Delta }{d})\frac{\pi d^{2}}{4},$$
where $G_{sp}$ denotes the shear modulus of the hardening zone of the material and is given by:(16)$$G_{sp}=\frac{1}{3}\frac{f_{u}-f_{y}}{\varepsilon_{u}-\varepsilon_{y}},$$
where $f_{u}$ is the ultimate strength, $f_{y}$ is the yielding strength, $\varepsilon_{u}$ is the ultimate strain, and $\varepsilon_{y}$ is the yielding strain of the stud. Consequently, the shear stiffness of the connector in stage 3 is:(17)$$K_{\mathrm{III}}=\frac{\pi d^{3}G_{sp}}{4\left(d^{2}+\Delta_{2}^{2}\right)}.$$

When the strain of the stud reaches the ultimate strain capacity $\varepsilon_{u}$, the slip reaches the ultimate slip capacity $\Delta_{3}$, and the shear connector fails.

## 3. Validation of the Trilinear Model and Discussions

### 3.1. Load–Slip Curves

To ensure the comparability between different test results and the accuracy of comparison between the test results and the present model, the selected specimens or tests for comparison must meet the following requirements: (1) The test should be conducted following the standard push-out test layout by Eurocode 4, including the component dimensions, the test layout, and the loading procedure. Alternatively, the tests can exclude the interference of irrelevant factors. (2) The failure mode of specimens can be confirmed to be dominated by the bending–shear failure of stud shanks according to the description in the literature. (3) The material and geometric parameters of the specimens required by the present model are clearly stated in the literature or can be calculated using the given parameters and relevant specifications.

Based on the above discussion, the test results of specimens QT1, QT2, and QT3 by Xu et al. [39]; GL19 by Guezouli & Lachal [40]; SP3-1, SP3-2, SP3-3, SP4-1, SP4-2, and SP4-3 by Okada et al. [5]; and ST25A, ST25B, ST27A, and ST30A by Shim et al. [41] are chosen to validate the present trilinear model. We also compare the existing codes and empirical formulae with the present analytical method and the test results. Numerous empirical load–slip formulae exist in which the fractional-type [13,32,42,43,44] and exponential-type [2,12,40,44,45,46,47,48,49] curves are widely accepted. However, the head stud shear connector’s shear strength must be determined before applying the empirical formulae. The shear strength formulae for headed studs shear connectors in the Chinese Code GB50917-2013 [11] and the European Code Eurocode 4 [12] are expressed as the following:(18)$$\begin{array}{l} V_{u1}=\min\left\{0.29\alpha d_{s}{}^{2}\sqrt{f_{cly}E_{c}},0.8A_{s}f_{u}\right\}\\ V_{u2}=\min\left\{0.43A_{s}\sqrt{f_{cu}E_{c}},3\lambda_{p}A_{s}{\left(\frac{E_{c}}{E_{s}}\right)}^{0.4}{\left(\frac{f_{cu}}{f_{u}}\right)}^{0.2}\right\} \end{array},$$
where:(19)$$\begin{array}{l} \lambda_{p}=\left\{\begin{array}{l} 6-\frac{\phi_{s}}{1.05},\;\phi s<5\\ 1,\;\;5\leq \phi_{s}\leq 7\\ \phi_{s}-6,\;\phi_{s}\geq 7 \end{array}\right.\\ \phi_{s}=h_{s}/d_{s}\\ \alpha =\min\left\{0.2\left(\phi_{s}+1\right),1\right\} \end{array},$$

Four existing load–slip curve formulae are selected for comparison with the present model, where $V_{1}$ and $V_{3}$ are fractional types, and $V_{2}$ and $V_{4}$ are the exponential types:(20)$$\left\{\begin{array}{l} V_{1}=V_{u1}\frac{2.24(s-0.058)}{1.98(s-0.058)+1}\\ V_{2}=V_{u1}\left(1-e^{-1.22s^{0.59}}\right)\\ V_{3}=V_{u2}\frac{s}{0.5+0.97s}\\ V_{4}=V_{u2}{\left(1-e^{-0.648s}\right)}^{\gamma } \end{array}\right.,$$
where:(21)$$\begin{array}{l} \gamma =-0.475\ln\left(\eta \right),\\ \eta =(1.0\times 10^{-5})E_{c}^{1.3}/d_{s}\leq 0.90; \end{array}$$
where *s* (mm) is the relative slip between the steel plate and the concrete plate; and *E* (MPa) and *d* (mm) represent the initial Young’s modulus of the concrete and the stud diameter, respectively.

For convenience, the specimens’ geometrical dimensions and material properties are given for the corresponding references and listed in Table 2.

The yielding and ultimate strains of the studs are assumed to be 0.002 and 0.1, respectively. The ultimate strain of specimen ST25A1-3 was taken as 0.34, as given in the literature [5]. The uniaxial compressive strains and nominal ultimate strains for different concrete strengths were selected as the recommended values in the Code for the Design of Concrete Structures (GB50010-2010) [50]. The spacings of the specimens QT1-2 are taken as their actual spacing, and the others are taken as *L* = 60 mm. The bearing capacity of the specimens QT1-3 enters a significant decline after the slip reaches 4mm, so $\Delta_{3}$ = 4 mm; the rest of the specimens are all taken as $\Delta_{3}$ = 6 mm.

The specimen QT’s stud diameter and concrete grade are 13 mm and C50, respectively. As shown in Figure 5, the initial stiffness and shear strength of the headed stud predicted using the existing empirical formulae are significantly lower than the test results. The traditional empirical formulae are based on the results of fitting experimental data; however, there are relatively few studies on the 13 mm diameter peg connectors. The lack of data leads the empirical fitting formula to a worse prediction of the shear behavior for the connectors with a stud diameter of 13 mm. The failure mode of small-diameter stud connectors differs significantly from that of commonly used stud connectors, which is primarily the shear failure of studs. The stud deformation contains too much shear deformation and almost no bending deformation, which can not sufficiently exploit the strength of the stud, leading the shear resistance weak. Apparently, the traditional empirical formula cannot reflect the above physical mechanism. The predicted results of the proposed analytical model are in high agreement with the experimental results, which indicates that the analytical model based on the foundation beam model and the deformation assumptions accurately reflects the interaction mechanism between the studs and concrete.

The stud diameter and concrete grade of the GL19 specimen are 19 mm and C55, respectively. As shown in Figure 6, the prediction accuracy of the existing empirical formulae for the headed stud shear connector with a diameter of 19 mm is improved compared to the connectors with a stud diameter of 13 mm. Since the stud failure mode for the studs with 19 mm is dominated by bending–shear damage, which can provide more shear load capacity than those with a diameter of 13 mm, they are relatively more commonly applied in engineering practices and test studies. Sufficient test data enhances the accuracy of the empirical fitting formulae. However, the empirical formulae are too conservative in predicting the initial stiffness of headed studs within a relative slip of 0–1 mm. The analytical model agrees well with the experimental results but underestimates the third-stage shear stiffness. The yield and tensile strengths of the studs selected in the specimen are significantly higher than ordinary stud materials, which, however, are not presented in the literature and are subsequently set to the same strain as the ordinary materials. Therefore, we consider the approximation of the strains as the reason for the prediction errors at this stage.

Figure 7 and Figure 8 show the load–slip curves for studs with a diameter of 22 mm embedded in C50 and C60 concrete, respectively. The present model accurately predicts the head stud shear connectors’ load–slip curves. In contrast, the results of the existing empirical formulae differ significantly from the test results. The traditional load–slip curve empirical formulas need to determine the shear strength of the connectors in advance, while the shear strength in the existing codes can only examine two specific cases: stud shear failure and concrete failure, and cannot reflect the difference between the strength ratio of the stud and concrete and the connector deformation against the impact of shear strength. The high compressive strength of the concrete used in the above specimens ensures that the concrete slab is barely damaged, and the stud deformation is sufficient, leading to a typical bending shear failure mode that differs from the shear damage mode investigated in the specifications. As a result, the test results of the shear strength of the stud connectors are about 30% higher than the predicted results of the empirical formula. The present analytical model, however, is based on the stiffness degradation of the stud’s bending and the concrete’s equivalent foundation, which reasonably examine the effect of the interaction between the stud and the concrete against shear behavior and break through the limitations of the traditional empirical fitting model.

The headed stud shear connector with a stud diameter of 25 mm can withstand a large shear load capacity and is generally regarded as a large connector. Due to few practical engineering applications and related research, 25 mm is usually listed as the upper limit of the applicable stud diameter in the specification for the empirical formula [12]. The load–slip curves of the existing tests with a diameter of 25mm were selected to compare with the results of the present analytical model and existing empirical equations where the concrete grades in Figure 9 and Figure 10 are C35 and C45, respectively. The test results in Figure 9 are discrete in the third stage since the low compressive strength of the C35 concrete cannot provide sufficient wrapping and support for the stud with a diameter of 25 mm, leaving the failure mode of the studs lying between concrete failure and studs shank failure. For intuitive purposes, the test data is averaged in the Figure. The empirical fitting and present analytical models provide satisfactory predictions for the specimens with concrete of C35. For the C45 specimen, the empirical result is too conservative in predicting the specimen’s stiffness. However, the present model is in excellent agreement with the test results, indicating that the analytical model is more applicable than the empirical fitting formula.

The prediction result of the present analytical model for the headed stud shear connectors with a diameter of 27 mm is shown in Figure 11. The present model precisely predicts the first two stages of the load–slip curve. The test results of stage 3 have a large discrepancy and, therefore, are averaged in blue. The maximum prediction error of the proposed analytical model in this stage is −8.7%. The empirical formula in the specification no longer applies to the connectors with stud diameters larger than 25 mm, so the empirical results are not shown in the Figure.

Figure 12 displays the load–slip curves of a group of headed stud shear connectors with a diameter of 30 mm, and the three replicate test results show large dispersions in the second and third stages. ST30A1 begins losing shear capacity at a relative slip of 3.1 mm, which is much less than the ductility requirement of the Eurocode (6 mm). The shear strength of the specimen is only 140 kN and even less than the yield strength of the stud. Although the shear behavior of specimens ST30A2 and ST30A3 is more potent than ST30A1, the concrete with a compressive strength of 40 MPa has an inadequate wrapping of the 30 mm diameter stud, resulting in extensive cracking and premature failure of the concrete, which still cannot make full use of the stud strength. This failure mode is quite different from the theoretical assumptions of this paper, so the proposed model can reasonably predict the shear behavior of the first stage when the concrete does not show significant damage. However, the prediction results of shear bearing capacity for the second and third stages are on the high side, with a maximum deviation of 20%. It should be emphasized that connectors with concrete failure have low shear strength and usually cannot meet the ductility requirements. Such non-ideal damage modes must be avoided as much as possible in engineering practice. In addition, for the headed stud connectors with a failure mode controlled by the concrete, the present theoretical formula should be appropriately modified.

### 3.2. Initial Shear Stiffness

The initial shear stiffness of the shear connectors significantly influences the mechanical behavior of the composite structures; therefore, this is also an important design aspect for the composite structures. Many codes define the initial stiffness of the headed stud connector. For example, the Chinese Code, Steel Structure Design Standard [51], specified the head stud shear connectors’ initial stiffness as the secant stiffness where the shear force reaches the shear strength $V_{u}$. The European Code Eurocode 4 [29] and Japanese Code JSSC [52] select the secant stiffness measured at $0.7V_{u}$ and $1/3V_{u}$ as the initial shear stiffness, respectively. The specification approaches generally define the secant stiffness at specific characteristic points on the load–slip curve as the head stud shear connectors’ initial stiffness, which are essentially a feature description of the existing curves. The established code approaches differ significantly from each other, require obtaining load–slip curves in advance, and cannot be directly derived from the calculation of the material and geometric parameters of the connectors, which results in severely restricted applicability.

The proposed model overcomes this limitation effectively. We compare the results of the proposed model and empirical formulae with the initial shear stiffness of the test, as shown in Table 3. The secant stiffness at the relative slip of 0.8 mm is chosen as the initial stiffness. It can be seen that the analytical model not only has a broad application range but also has significantly less prediction error than those of the empirical formulae. However, the errors and dispersion of the existing empirical formulae results are too much to reasonably predict the initial shear stiffness.

## 4. Conclusions

A new trilinear model is proposed to describe the load–slip behaviors of headed stud shear connectors regarding the load–slip curves and failure mode in the existing experimental results. Firstly, based on the beams on Winkler foundation and the principle of deformation energy equivalence, the analytical equations of shear stiffness for the first stages are derived. Then the equation for the equivalent stiffness of the third stage was established by assuming the stud shear failure mode and material elastoplasticity theory so as to obtain the analytical calculation model of the load–slip curve, which is verified by comparing the analytical model with the existing test results.

The conventional approach classifies the failure mode of the headed stud shear connector into two failure modes: stud failure and concrete failure, and provides the corresponding shear strength formulae. Then the load–slip curve is obtained when taking the shear strength into the shape-fitting formula. In other words, once the form of the empirical formula for the load–slip curve is selected, the shear behavior of the connectors is uniquely determined by the shear strength (peg diameter and strength or concrete compressive strength), which is inconsistent with physical reality. The proposed analytical approach comprehensively investigates the effects of material and geometric parameters of the stud and concrete, as well as the material degradation, on the shear behavior of connectors, which reflects the interaction mechanism between the stud and concrete. In particular, the load–slip curves of shear joints can be directly derived by incorporating the stud diameter, stud length, and the elastic-to-plastic institutive relation of the stud and concrete.

The proposed model remedies the shortcomings of the traditional empirical formulae, featuring higher accuracy, a more comprehensive range of parameter applicability, and definite physical significance, and lays the foundation for theoretical analysis of the interaction between the stud and concrete. However, it is noted that the present model applies to shear connectors with the failure mode of the stud bending and shearing. The applicability needs to be further validated and modified for the connectors with failure dominated by the pure shear damage of studs embedded in UHPC or the stud pull-out damage due to insufficient concrete wrapping.

## Figures and Tables

**Figure 1 materials-16-01173-f001:**
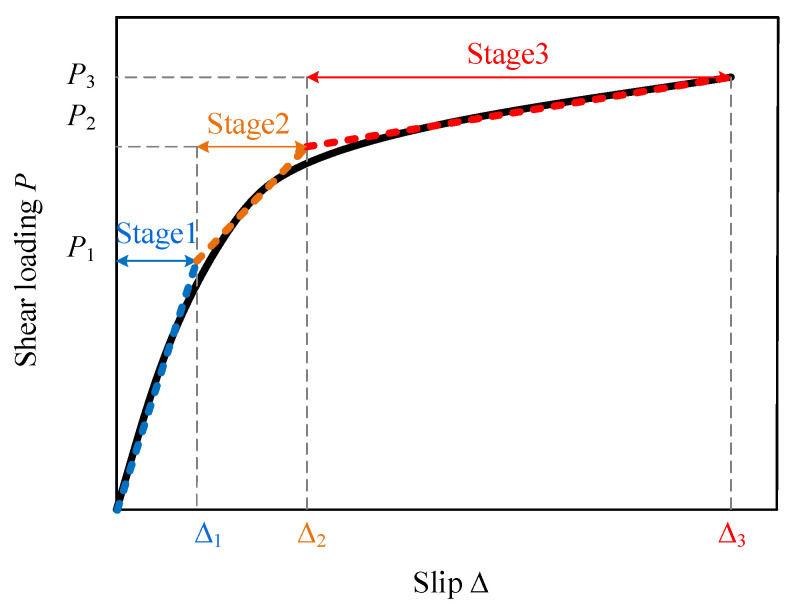
The typical load–slip curve and its approximation with triple segments.

**Figure 2 materials-16-01173-f002:**
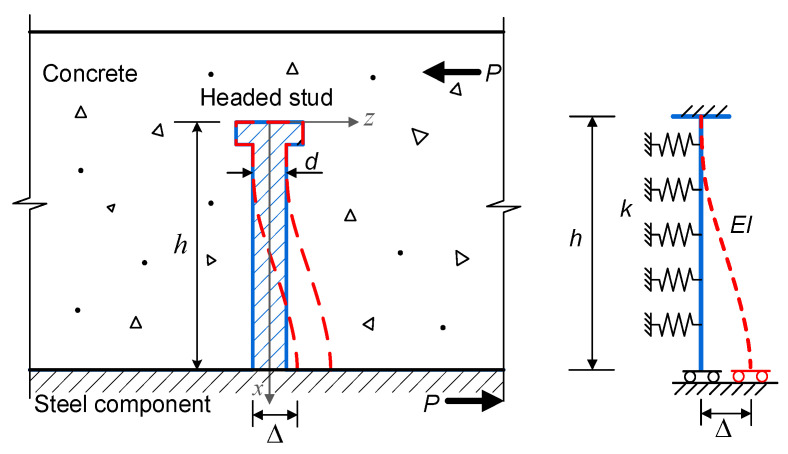
The deformation and its analytical model of the headed stud at stages 1 and 2.

**Figure 3 materials-16-01173-f003:**
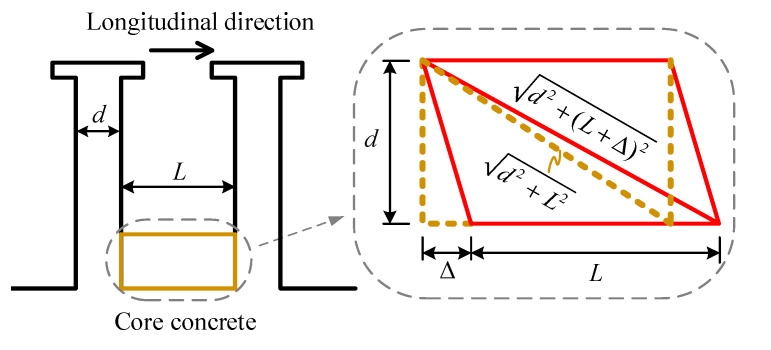
The deformation of the surrounding concrete at the stud’s root.

**Figure 4 materials-16-01173-f004:**
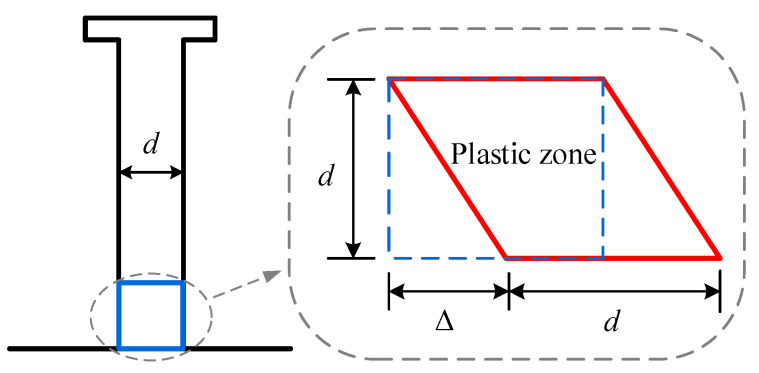
The plastic zone of the stud at its root.

**Figure 5 materials-16-01173-f005:**
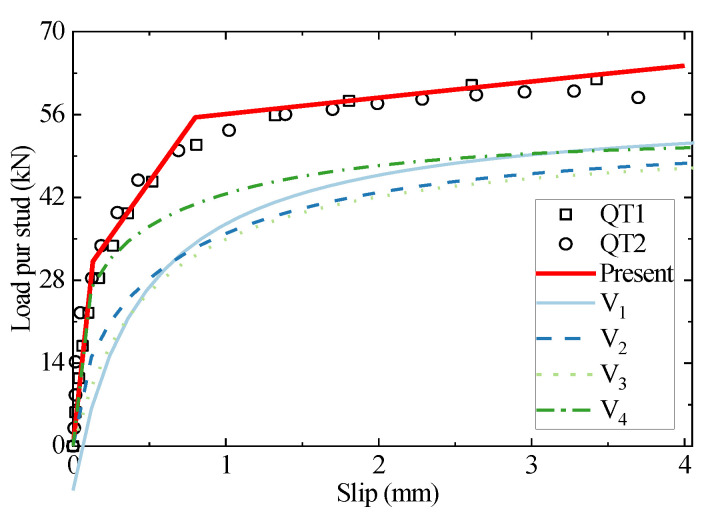
The comparison of load–slip curves between the test, empirical formulae, and the analytical method for QT1-2.

**Figure 6 materials-16-01173-f006:**
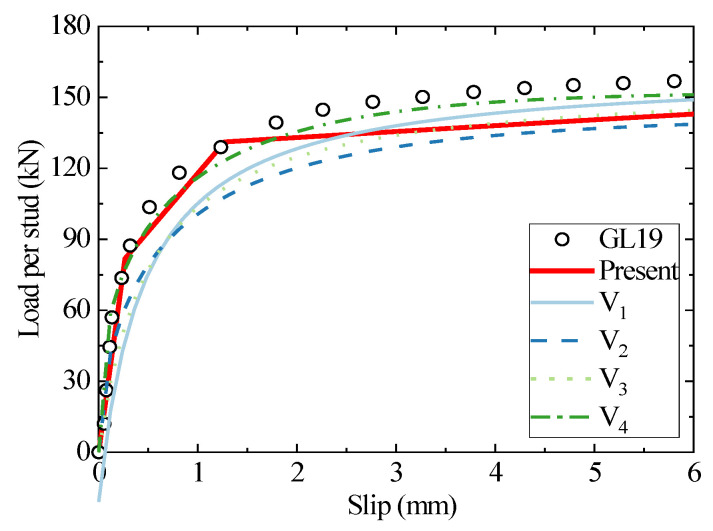
The comparison of load–slip curves between the test, empirical formulae, and the analytical method for GL19.

**Figure 7 materials-16-01173-f007:**
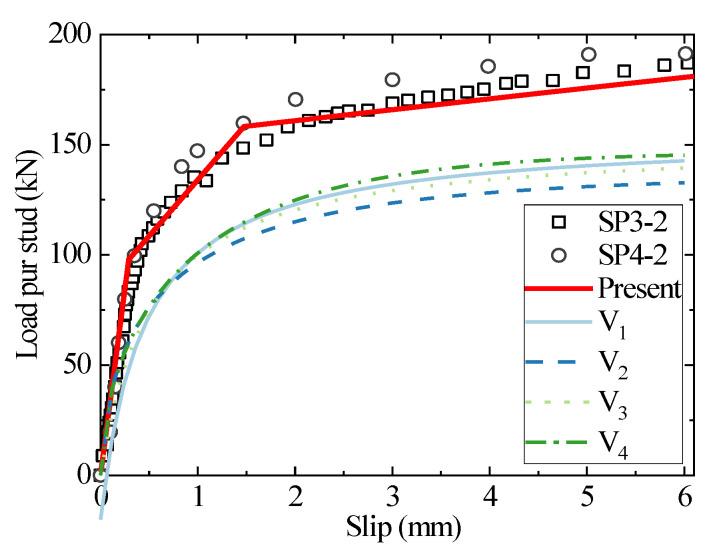
The comparison of load–slip curves between the test, empirical formulae, and the analytical method for SP3/4-2.

**Figure 8 materials-16-01173-f008:**
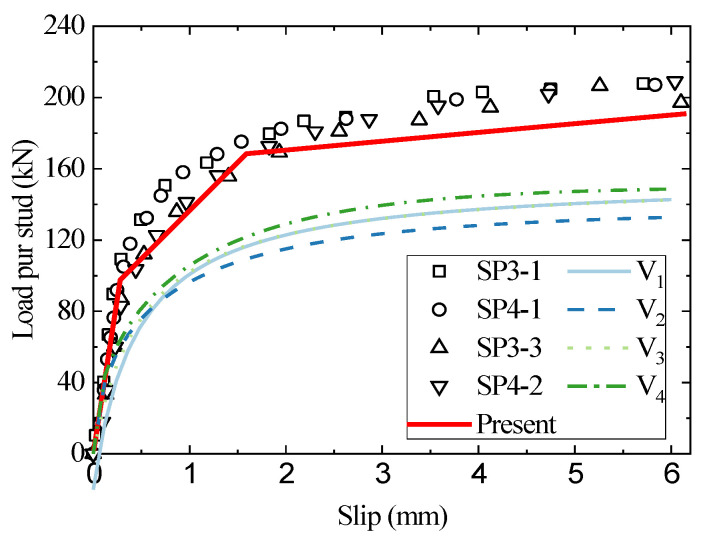
The comparison of load–slip curves between the test, empirical formulae, and the analytical method for SP3/4-1/3.

**Figure 9 materials-16-01173-f009:**
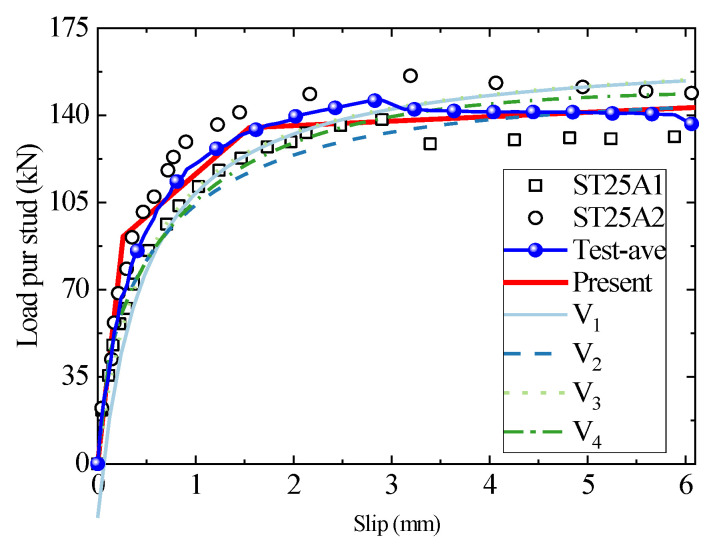
The comparison of load–slip curves between the test, empirical formulae, and the analytical method for ST25A1/2.

**Figure 10 materials-16-01173-f010:**
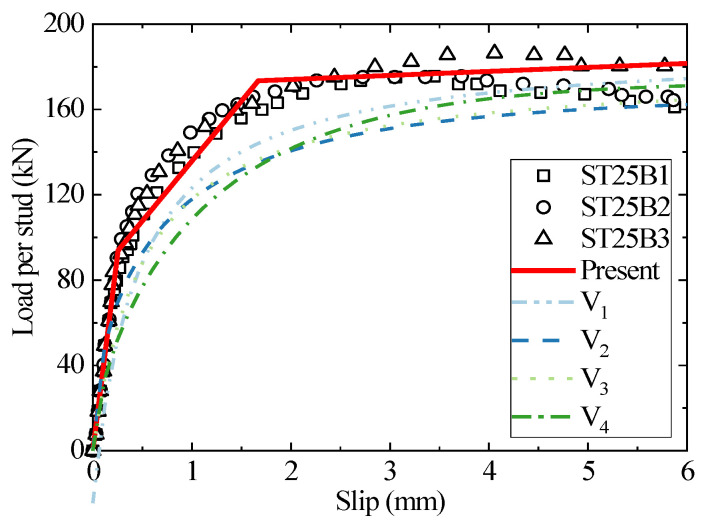
The comparison of load–slip curves between the test, empirical formulae, and the analytical method for ST25B1-3.

**Figure 11 materials-16-01173-f011:**
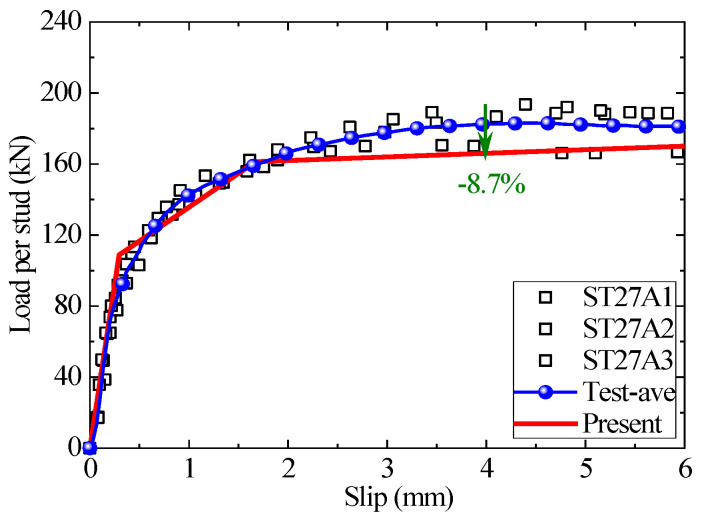
The comparison of load–slip curves between the test, empirical formulae, and the analytical method for ST27A1-3.

**Figure 12 materials-16-01173-f012:**
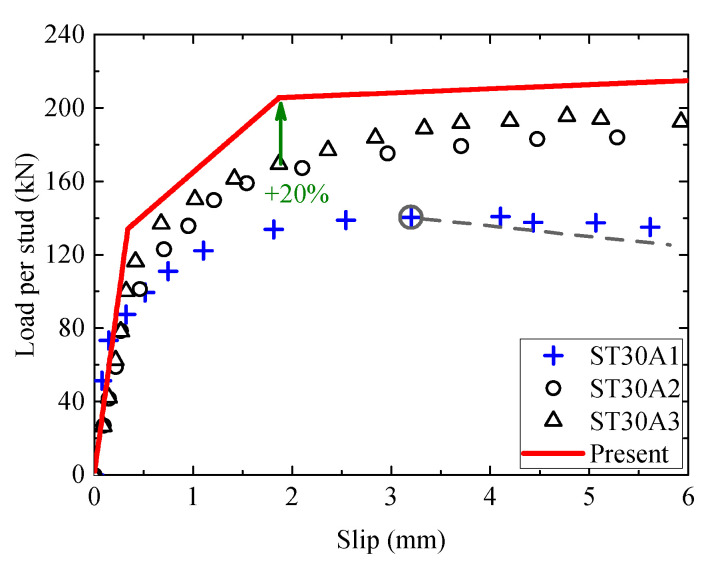
The comparison of load–slip curves between the test and the analytical method for ST30A1-3.

**Table 1 materials-16-01173-t001:** Comparison of the concrete plastic zone.

No.	Specimen	D (mm)	f_cly_ (MPa)	E_c_ (GPa)	R_c_ (mm)	X_m_ (mm)	R_c_/d	X_m_/d
1	D13-C40	13	40	34.56	12.50	13.13	0.96	1.01
2	D13-C32	13	32	32.64	13.60	13.31	1.05	1.02
3	D13-C24	13	24	29.87	14.20	13.61	1.09	1.05
4	D16-C40	16	40	34.56	15.50	16.58	0.97	1.04
5	D16-C32	16	32	32.64	16.80	16.82	1.05	1.05
6	D16-C24	16	24	29.87	18.10	17.19	1.13	1.07
7	D19-C40	19	40	34.56	19.00	20.12	1.00	1.06
8	D19-C32	19	32	32.64	20.00	20.40	1.05	1.07
9	D19-C24	19	24	29.87	21.80	20.86	1.15	1.10
10	D22-C40	22	40	34.56	22.60	23.72	1.03	1.08
11	D22-C32	22	32	32.64	25.00	24.06	1.14	1.09
12	D22-C24	22	24	29.87	26.00	24.60	1.18	1.12

d, f_cly_, and E_c_ represent stud diameter, compressive strength of concrete, and Young’s Modulus of concrete, respectively; X_m_ and R_c_ are the effective concrete plastic zone in Meng et al. [35] and Xu et al. [38], respectively.

**Table 2 materials-16-01173-t002:** The parameters of the experimental specimen.

Parameters	QT1 QT2 QT3	GL19	SP3-1 SP3-3 SP4-1 SP4-3	SP3-2 SP4-2	ST25A1 ST25A2 ST25A3	ST25B1 ST25B2 ST25B3	ST27A1 ST27A2 ST25A3	ST30A1 ST30A2 ST30A3
*d* (mm)	13	19	22	22	25	25	27	30
*h* (mm)	80	100	150	150	155	155	155	155
*f_y_* (MPa)	400	500	445	445	328	328	328	328
*f_u_* (MPa)	480	530	530	530	426	426	426	426
*f_cu_* (MPa)	50	50	60	50	40	50	40	40
*ε_y_*	0.002	0.002	0.002	0.002	0.002	0.002	0.002	0.002
*ε_u_*	0.1	0.1	0.1	0.1	0.34	0.34	0.34	0.34
*E_s_* (Gpa)	200	210	210	210	213	213	213	213
*L* (mm)	60	100	110	110	250	250	250	250

**Table 3 materials-16-01173-t003:** The comparison of initial shear stiffness between test results, existing formulae, and the present method.

Specimen	Tests	Equation (1)		Equation (2)		Equation (3)		Equation (4)		Present	
	K_test_ kN/mm	K_p1_ kN/mm	Error %	K_p2_ kN/mm	Error %	K_p3_ kN/mm	Error %	K_p4_ kN/mm	Error %	K_p_ kN/mm	Error %
QT1	63.4	42.9	−32.33	41.4	−34.7	40.0	−36.9	50.9	−19.7	68.0	7.3
QT2	64.3	42.9	−33.28	41.4	−35.6	40.0	−37.8	50.9	−20.8	68.0	5.8
GL19	145.8	121.9	−16.39	119.5	−18.0	117.1	−19.7	135.9	−6.8	137.6	−5.6
SP3-2	162.5	112.3	−30.89	116.5	−28.3	115.1	−29.2	115.8	−28.7	156.9	−3.4
SP4-2	172.8	112.3	−35.01	116.5	−32.6	115.1	−33.4	115.8	−33.0	156.9	−9.2
SP3-1	184.0	116.0	−36.96	112.7	−38.8	120.4	−34.6	123.8	−32.7	159.8	−13.2
SP4-1	181.5	116.0	−36.09	112.7	−37.9	120.4	−33.7	123.8	−31.8	159.8	−12.0
SP3-3	170.9	116.0	−32.12	112.7	−34.1	120.4	−29.5	123.8	−27.6	159.8	−6.5
SP4-3	166.4	116.0	−30.29	112.7	−32.3	120.4	−27.6	123.8	−25.6	159.8	−4.0
ST25A1	132.4	124.4	−6.04	124.6	−5.9	127.6	−3.6	121.6	−8.2	138.8	4.8
ST25A2	148.4	124.4	−16.17	124.6	−16.0	127.6	−14.0	121.6	−18.1	138.8	−6.5
ST25B1	163.4	143.5	−12.18	139.0	−14.9	136.1	−16.7	124.4	−23.9	159.0	−2.7
ST25B2	171.6	143.5	−16.38	139.0	−19.0	136.1	−20.7	124.4	−27.5	159.0	−7.3
ST25B3	176.4	143.5	−18.65	139.0	−21.2	136.1	−22.8	124.4	−29.5	159.0	−9.9
ST27A1	134.2	/	/	/	/	/	/	/	/	128.1	−4.5
ST27A2	136.5	/	/	/	/	/	/	/	/	128.1	−6.2
ST27A3	133.8	/	/	/	/	/	/	/	/	128.1	−4.3
ST30A1 *	86.5	/	/	/	/	/	/	/	/	87.5	1.2
ST30A2 *	82.3	/	/	/	/	/	/	/	/	87.5	6.3
ST30A2 *	91.4	/	/	/	/	/	/	/	/	87.5	−4.3
Averaged absolute error		25.20		26.38		25.73		23.84		6.24	

/ indicates that the existing empirical formula is not applicable here; * indicates the secant shear stiffness at the relative slip of 0.25 mm.

## Data Availability

Not applicable.

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
