# Peer review of "A Trilinear Model for the Load–Slip Behavior of Headed Stud Shear Connectors"

_materials, 2023, doi:10.3390/ma16031173_

Round 1

Reviewer 1 Report

The paper is interesting and clearly written. The methodology and results are clearly presented, and all results support the conclusions. I am of the opinion that the work can be published in the presented form.

Author Response

The authors appreciate your inspiring comments and recommendation to publish!

Reviewer 2 Report

In the manuscript “A trilinear model for the load-slip behavior of headed stud shear connectors”, Meng, Wang, and Xu propose a model to describe the load-slip behavior of heated stud shear connectors. They use a new trilinear model which describes the load-slip behavior of the headed stud shear connectors. The model is compared with results from the literature.

The paper is well-written and clear, so it may be published in its current form.

Author Response

The authors appreciate your close read and recommendation to publish!

Reviewer 3 Report

In this paper the authors proposed a trilinear model to investigate the load slip behaviour f headed stud shear connection in steel concrete structures. The analytical model presented take into account the foundation stiffness by an empirical correlation based on authors previous work. In the present paper the authors proposed a numerical procedure able to reproduce the experimental tests. The paper is well organised and the topic is of interest for Materials readers.

Few comments to the authors are listed below:

1.       The authors claimed a new trilinear model used for describe the load slip behaviour of headed stud connections. Could the authors compare their model with other available in literature? (e.g. Xu, X.; Zeng, S.; He, W.; Hou, Z.; He, D.; Yang, T. Numerical Study on the Tensile Performance of Headed Stud Shear Connectors with Head-Sectional Damage. Materials 2022, 15, 2802. https://doi.org/10.3390/ma15082802). Please revise the introduction section reporting relevant papers.

2.         In the section 2.3 the authors present three empirical equations for describe the equivalent stiffness of the foundation. Does the authors investigate the difference in predictions by all the models? If their work started directly by their previous work (reference 29) the section is superfluous.

 The paper could be considered for publication after minor revisions.

Reviewer 4 Report

The paper is scientifically sound and well presented. 

Congratulations on a very concise and well written review of literature. The foundation of this literature review presents the soundness the researchers provided to the work. Couple of very minimal edits required are all semantic. 

1. Please have a thorough review of grammar, as there are a few recurring grammatical errors. 

2. Please use better math formatting for formulae. Some of the equations are troublesome to understand at first glance due to in-line formatting of math equations (Eqn 4,5 etc).

3. Please provide a background on how the deformation for Winkler foundation is obtained in closed form for equation 1. 

Reviewer 5 Report

The manuscript is large and detailed study of the load-slip behavior of headed stud shear connectors using analytical methods (trilinear model). The article is carefully designed, accumulates a large amount of data and shows interesting and significant results, however, as authors mentioned, the applicability of the results needs to be further validated. I would recommend to accept the manuscript after minor revision according to the following commnet:

Figures 5-12 show comparison of calculated curves with experiments, but more accurate and clear references on using experimental results are needed. Now it’s hard to understand from which source the experimental data was chosen and how experiments were conducted for different specimens.

Reviewer 6 Report

In this paper, a new derivation method of load-slip curves (P vs Δ) for headed stud shear connectors was studied and shown to be in good agreement with experimental results. This paper contains sufficient content and depth. 

Please consider the following points.

(1) Figure 4: In this paper, the height of the concrete plastic zone is assumed to be d. Please consider the effect of this assumption on P vs Δ relation.

(2) Is the plastic hardening rule for headed studs (steel) linear hardening or is it using the power law hardening?

(3) Section 2.5 (Stage 3): What is the interfacial strength or peel strength at the interface between concrete and stud (steel)? In Figures 6, 8 and 11, at stage 3, the difference between the experimental value and the predicted value is large. Is this difference related to the interfacial strength mentioned above?

(4)Line 267 should be as follows?

"The yielding and ultimete strains of the studs are assumed to be 0.002 and 0.1, respectively"?
